# Self-consistent dynamical maps for open quantum systems

Orazio Scarlatella[1,2,3]⋆ and Marco Schirò[1]

**1** JEIP, UAR 3573 CNRS, Collège de France, PSL Research University,
11 Place Marcelin Berthelot, 75321 Paris Cedex 05, France
**2** Pritzker School of Molecular Engineering, University of Chicago,
5640 South Ellis Avenue, Chicago, Illinois 60637, U.S.A.
**3** T.C.M. Group, Cavendish Laboratory, J.J. Thomson Avenue, Cambridge CB3 0HE, UK

⋆ os444@cam.ac.uk

## Abstract

In several cases, open quantum systems can be successfully described using master equations relying on Born-Markov approximations, but going beyond these approaches has become often necessary. In this work, we introduce the NCA and NCA-Markov dynamical maps for open quantum systems, which go beyond these master equations replacing the Born approximation with a self-consistent approximation, known as non-crossing approximation (NCA). These maps are formally similar to master equations, but allow to capture non-perturbative effects of the environment at a moderate extra numerical cost. To demonstrate their capabilities, we apply them to the spin-boson model at zero temperature for both a Ohmic and a sub-Ohmic environment, showing that they can both qualitatively capture its strong-coupling behaviour, and be quantitatively correct beyond standard master equations.



# 1  Introduction

The theory of open quantum systems, born to describe nuclear magnetic resonance (NMR) [1–3] and lasers [4,5], is now of fundamental importance for the development of quantum devices [6–11], to describe chemical reactions [12–15], understand biological complexes [16–21] and to explore novel non-equilibrium quantum states [22–25].

For typical quantum optical systems, where the coupling with the environment is weak and the environment is unstructured, one can rely on Born-Markov master equations [2,26–35] and on equivalent stochastic approaches [36–38]. Nevertheless, nowadays going beyond these approaches is necessary for a growing number of cases, including electronic transport problems [39–43], optomechanical resonators [44], quantum dots [45,46], superconducting circuits [47–51] and quantum simulation platforms [52–57] where non-Markovian and non-perturbative phenomena are expected [55,58–62].

Theoretical approaches beyond Born-Markov master equations have recently been developed, including phenomenological master equations [39–41,63–65], but also microscopic approaches such as diagrammatic techniques [66–72] including diagrammatic Monte-Carlo [73–78], hierarchies of exact equations of motion [79–86] and matrix product states approaches [87,88]. The latter microscopic techniques can be very powerful, but they are often computationally demanding and formally more involved than Born-Markov master equations, limiting their applicability in many cases.

On the other hand, a powerful technique of many-body theory is performing partial summations of perturbation series, such as in several textbook applications [89,90]. This allows to obtain analytical equations that, despite being simple, can capture complex non-perturbative phenomena. An example are the non-crossing approximations (NCA), that have been used for example for disordered systems [89], for quantum impurity models, both in and out of equilibrium [91–95], and for quantum transport problems [96–99]. In [100] a NCA has been formulated to conveniently capture the dynamics of quantum systems coupled simultaneously to a Markovian and to a non-Markovian environment.

In this work, we show how for generic open quantum systems the non-crossing approximation of [100] can be used to go beyond the Born approximation, underlying standard master equations such as the Redfield [2] or Lindblad-Davies [27,28], and that it can also be combined with a usual Markovian approximation. The result is an equation for the dynamical map propagating the system density matrix, rather than for the density matrix itself, that we call

the NCA or NCA-Markov map, which is formally very similar and reduces to the usual Born and Born-Markov master equations at sufficiently weak coupling, but has a *self-consistent* dissipator allowing to capture non-perturbative effects at a moderate extra numerical cost. Furthermore, we discuss how the NCA maps can be benchmarked evaluating the leading-order self-consistent correction, the one crossing approximation (OCA).

We then apply the NCA and NCA-Markov maps to the spin-boson model with a zero temperature environment. We show that in the case of a Ohmic environment both approaches correctly capture the non-perturbative dynamics of this model at strong system-bath coupling, including its crossover from coherent to incoherent dynamics and its quantum phase transition to a localized phase, which are completely missed by the Born and Born-Markov master equations. Finally, computing the one-crossing correction to NCA we show for both a Ohmic and a sub-Ohmic environment that these approaches are quantitatively accurate in regimes in which the latter master equations display significant deviations.

## 2 From master equations to dynamical maps

We consider a generic quantum system coupled to a bath with total Hamiltonian $H_{\text{tot}} = H_S + H_B + H_{SB}$, where $H_S$ and $H_B$ are the system and bath Hamiltonians. The main assumptions behind the self-consistent maps are that the bath is described by a set of non-interacting bosonic or fermionic modes and that the system-bath density matrix factorizes at time $t = 0$, $\rho_{\text{tot}}(0) = \rho(0) \otimes \rho_B(0)$. To introduce the method, we restrict here to a single bath of bosonic modes, with annihilation operators $a_i$ and Hamiltonian $H_B = \sum_i \omega_i a_i^\dagger a_i$, and to a system-bath coupling of the form $H_{SB} = X \otimes B$, respectively with system and bath operators $X = X^\dagger$ and $B = \sum_i \frac{\lambda_i}{2}(a_i + a_i^\dagger) = B^\dagger$ and coupling strength $\lambda_i$. We also consider an initial bath density matrix $\rho_B(0)$ that is stationary with respect to $H_B$. We remark though that the maps apply in more general cases, as we discuss in Sec. 2.2.

Under those assumptions, the reduced dynamics of the system is often conveniently described by master equations of the form [101]

$$\partial_t \rho(t) = \hat{\mathcal{H}}_S \rho(t) + \int_0^t dt_1 \hat{\mathcal{D}}(t - t_1)\rho(t_1), \tag{1}$$

where $\hat{\mathcal{D}}$ is the dissipator, describing the influence of the bath on the system. Weak-coupling master equations are based on a second-order approximation in the system-bath coupling, the Born approximation [101], that leads to the dissipator

$$\hat{\mathcal{D}}_{\text{Born}}(\tau) = \Gamma(\tau)\left(e^{\hat{\mathcal{H}}_s \tau}[X\bullet]X - X e^{\hat{\mathcal{H}}_s \tau}X\bullet\right) + \text{H.c.}, \tag{2}$$

in which the superoperator $\hat{\mathcal{H}}_s \bullet = -i[H_S, \bullet]$ describes the bare system dynamics and where $\Gamma$ is the bath correlation function

$$\Gamma(\tau) = \text{tr}[B(\tau)B(0)\rho_B(0)], \tag{3}$$

with $B(\tau)$ time-evolved with the bath Hamiltonian $H_B$. We used square brakets in (2) when necessary to indicate the argument of superoperators. We refer to (1), (2) as the Born master equation. The dissipator (2) has the familiar structure of Redfield [2] and Lindblad [28] master equations, as it is the starting point for deriving them.

Heuristically, we introduce the NCA self-consistent map in two steps: first we recognize that the dynamical map $\hat{\mathcal{V}}(t)$, namely the superoperator evolving the system density matrix

Figure 1: The expression at the top represents the (first terms of the) exact series for the dissipator $\hat{\mathcal{D}}$ which is the self-energy of the Dyson equation (4): non-bold solid lines represent bare evolution superoperators $e^{\hat{\mathcal{H}}_s(\tau)}$, dashed lines correspond to bath correlation functions (3), with a typical decay time $\tau_b$. The middle expression shows that retaining the 1st term of the series corresponds to the Born dissipator (2). At the bottom, the NCA dissipator (5) corresponds to the sum of all the diagrams in which the dashed lines do not cross (labelled by "NC"), which can be expressed in terms of a bold solid line representing the dynamical map $\hat{\mathcal{V}}$. Similarly, $\hat{\mathcal{D}}_{\mathrm{OCA}}$ is the dissipator including the leading-order self-consistent correction to NCA, namely in the one-crossing approximation (OCA). We also show that $\hat{\mathcal{D}}_{\mathrm{OCA}}$ decays on a longer time-scale $2\tau_b$ than $\hat{\mathcal{D}}_{\mathrm{NCA}}$, decaying in $\tau_b$.

from the initial state, $\rho(t) = \hat{\mathcal{V}}(t)\rho(0)$, obeys an analogous equation:

$$\partial_t \hat{\mathcal{V}}(t) = \hat{\mathcal{H}}_S \hat{\mathcal{V}}(t) + \int_0^t dt_1 \hat{\mathcal{D}}(t - t_1)\hat{\mathcal{V}}(t_1). \tag{4}$$

Note that such a map can always be defined, and (4) only relies on the two main assumptions outlined above.

Then, we make the Born dissipator self-consistent, by replacing in (2) the evolution super-operator $e^{\hat{\mathcal{H}}_s \tau}$, describing the dynamics of the system isolated from the environment, with the dynamical map $\hat{\mathcal{V}}(\tau)$ itself, taking into account the influence of the environment:

$$\hat{\mathcal{D}}_{\mathrm{NCA}}(\tau) = \Gamma(\tau)\big(\hat{\mathcal{V}}(\tau)[X\bullet]X - X\hat{\mathcal{V}}(\tau)X\bullet\big) + \mathrm{H.c.}, \tag{5}$$

where for taking the Hermitian conjugate one can use the property $(\hat{\mathcal{V}}[\bullet])^\dagger = \hat{\mathcal{V}}[\bullet^\dagger]$. Eq. (4) with the dissipator (5) has a very similar structure to standard master equations, but is an equation for the dynamical map rather than for the density matrix. We call its solution the *NCA dynamical map*. Differently from standard master equations, the dissipator (5) is determined *self-consistently* as it depends on the unknown. Note also that *non-Markovian effects* are captured both by the time-integral in Eq. (4) and by the self-consistent dissipator, coupling the map to its values at earlier times.

Apart from this heuristic derivation, a formal derivation using diagrammatic techniques is given in App. C, where (4) is recognized to be the Dyson equation for the perturbation theory of the dynamical map $\hat{\mathcal{V}}(t)$ in the system-bath coupling and the dissipator $\hat{\mathcal{D}}$ to be its self-energy (the same is true for Eq. (1) and $\rho(t)$). An important consequence is that (4) (as (1)) is in principle an exact equation for bosonic or fermionic environments, where the exact self-energy $\hat{\mathcal{D}}$ is defined by its perturbation series. A diagrammatic representation of this series along with the approximations schemes considered in this manuscript is shown in Fig. 1: a lowest-order truncation of the series corresponds to the Born approximation (2), while approximating it with the sum of its infinitely-many "non-crossing" diagrams (hence the name NCA) one finds that the sum of the series has a compact expression in terms of $\hat{\mathcal{V}}(t)$, namely Eq. (5).

The NCA map is non-perturbative, as its dissipator (5) includes contributions to all orders in the bath correlation function $\Gamma$ (or in the system-bath coupling $H_{SB}$), in contrast with the Born dissipator (2) which is first order in $\Gamma$ (2nd order in $H_{SB}$). To see this explicitly, one can obtain $\hat{\mathcal{V}}$ by integrating Eq. (4) with dissipator (5), and plug it back in Eq. (5) to get an expression for the dissipator which explicitly depends on $\Gamma$, $\Gamma^2$ and $\hat{\mathcal{V}}$: substituting $\hat{\mathcal{V}}$ recursively, contributions up to all powers in $\Gamma$ are generated. This allows the NCA map to capture non-perturbative effects with respect to standard master equations based on the Born approximation, like Redfield [2] and Lindblad-Davies equations [27,28]. We also remark that, when the coupling is sufficiently weak the NCA maps become equivalent to standard weak-coupling master equations, as to lowest order in the system-bath coupling the NCA dissipator (5) reduces to the Born one (2): $\hat{\mathcal{V}}(\tau)$ in (5) reduces to $e^{\hat{\mathcal{H}}_s \tau}$.

A discussion of the regimes of validity of the NCA maps, of how to benchmark their predictions and of their applicability is reported in Section 2.2. In the following Section we show how the usual Markovian approximation done for master equations can be combined with the NCA, how a similar steady-state equation holds and also a quantum regression theorem for computing correlation functions.

## 2.1 Markovian approximation, steady-state and quantum regression theorem

When the bath-induced dynamics on the system is slower than the typical decay time of the dissipator, one can do a partial *Markovian approximation*, replacing $\hat{\mathcal{V}}(t_1)$ in (4) with $e^{\hat{\mathcal{H}}_s(t_1-t)}\hat{\mathcal{V}}(t)$, yielding:

$$\partial_t \hat{\mathcal{V}}(t) = \left[ \hat{\mathcal{H}}_S + \int_0^t d\tau \, \hat{\mathcal{D}}(\tau) \, e^{-\hat{\mathcal{H}}_s(\tau)} \right] \hat{\mathcal{V}}(t) \,. \tag{6}$$

We call the solution of (6) and (5) the NCA-Markov dynamical map. The same approximation is routinely done at the level of master equations (1), where $\rho(t_1)$ is replaced with $e^{\hat{\mathcal{H}}_s(t_1-t)}\rho(t)$, leading to what we will refer to as the Born-Markov master equation, that we will use for comparison. The NCA-Markov has a computational advantage over the NCA, as the integral in (6) can be updated at each time-step, and does not need to be recomputed entirely as in (4), leading to a reduced numerical cost as we will discuss in Sec. 2.3. We remark that this approximation does not lead to a fully Markovian equation. This is already true for the Born-Markov equation we will compare with, where the finite upper time-integration limit, analogously as in (6), introduces some non-Markovian effects and results in a more accurate approach [102] (see also Appendix D). The NCA-Markov map captures even further non-Markovian effects, as it still depends explicitly on its previous times through the self-consistent dissipator (5). We further discuss the validity of the NCA-Markov approximation in App. C.4.

For master equations, the *steady-state* can be computed by the the generator of their infinitesimal time-evolution, the Liouvillian, by an algebraic equation. An analogus expression exists here. Assuming that the system forgets its initial conditions at a sufficiently long time and reaches a time-independent steady-state, defined as $\rho_s = \lim_{t\to\infty} \hat{\mathcal{V}}(t,0)\rho(0)$, this state also obeys the equation [100, 103]:

$$\left( \hat{\mathcal{H}}_S + \int_0^\infty d\tau \, \hat{\mathcal{D}}(\tau) \right) \rho_s = 0 \,. \tag{7}$$

We remark that at weak system-bath coupling $\hat{\mathcal{D}}(\tau)$ is expected to decay on a much shorter timescale than that for the system to reach the steady-state. Eq. (7) then allows to extract $\rho_s$ from the short-time dynamics of the system, akin to the condition for Markovian master equations.

*The quantum regression theorem* for Markovian master equations allows to compute (multi-time) correlation functions from the same generator of the dynamics which evolves the density matrix [101]. This generalizes to NCA, for which it can be proven without making a Markovian assumption [103]. Take a generic system operator $Y$, then

$$\langle Y(t)Y^{\dagger}(t')\rangle = \mathrm{tr}\big[Y\hat{\mathcal{V}}(t-t')Y^{\dagger}\rho(t')\big]\theta(t-t') + \mathrm{tr}\big\{Y^{\dagger}\hat{\mathcal{V}}(t'-t)[\rho(t)Y]\big\}\theta(t'-t).$$

(8)

Note that $\langle Y(t)Y^{\dagger}(t')\rangle = \langle Y(t')Y^{\dagger}(t)\rangle^*$ relates the $t > t'$ with the $t < t'$ expression. We remark that a similar result is not expected to hold for generic non-Markovian approaches and it is a peculiar property of NCA. Note that the numerical cost for evaluating these correlation functions in NCA is the same as for computing $\hat{\mathcal{V}}$, making correlation functions easily accessible. An application to the spin-boson model is given in Sec. E.1.

## 2.2 Validity and generalizations

The NCA maps can be generalized to a whole hierarchy of self-consistent approximations, where the dissipators always depend on $\hat{\mathcal{V}}$ rather than on $e^{\hat{\mathcal{H}}_s\tau}$. This is achieved using standard results of many-body theory that allow to rewrite self-energies in terms of *skeleton* diagrams [104]. Each approximation in the hierarchy has contributions up to all powers in the system-bath coupling in the dissipator, still the hierarchy is ordered, such that higher-order terms become negligible for a sufficiently small system-bath coupling. The NCA corresponds to lowest order approximation of the hierarchy.

The NCA maps therefore become quantitatively correct when the coupling with the environment is sufficiently weak, such that the higher-order self-consistent corrections become negligible. We remark that they might still be quantitatively accurate in regimes in which Born-based master equations already show significant deviations from the correct solution, as we will demonstrate for the spin-boson model in Sec. 3.2. Given their non-perturbative nature, the NCA maps might also be able to capture qualitatively the physics in the strong-coupling regime, as we show in Sec. 3, while quantitative accuracy is not guaranteed there.

The natural strategy to assess the validity of the results of the NCA maps is to evaluate the leading-order self-consistent correction. This is known as "one-crossing approximation" (OCA) (see e.g. [94,105]) and still has a compact expression adding one term to the NCA dissipator, that we derive and report in Appendix C.5. We compute and discuss the OCA corrections for the spin-boson model in Sec. 3.2. Note also that going one order beyond OCA is feasible [94,106] and a Monte-Carlo sampling around NCA is also possible [75,105].

Compared to master equations, we remark that the decay time $\tau_b$ of the bath correlation function $\Gamma(\tau)$ is the shortest timescale for system-bath interactions in those equations [33, 101,107–109], while the NCA maps capture processes that happen on faster scales. Also, for both NCA and Born approximations, $\tau_b$ is the decay time of the dissipator and therefore the timescale over which Eq. (4) loses its "memory" of the past. Upon including higher-order self-consistent diagrams, the decay-time of the dissipator and thus the memory allowed by these approximations systematically increases; this is also shown in Fig. 1. Also remark that the NCA maps exactly preserve trace and Hermiticity of the density matrix [100].

While so far we focused on a single bath of bosonic variables coupled to the system with their real part, the NCA maps generalize to multiple baths, either Markovian or treated at the same level of approximation, and extend to complex couplings such as in the rotating-wave approximation [100], for which we report the dissipator in Appendix B. Also, these maps and their higher-order counterparts naturally generalize to fermionic environments [100], to non-stationary environments [103] and to driven systems (time-dependent Hamiltonians). Instead, their derivation does not extend to spin environments, for which the Wick's theorem does not apply. Nevertheless, they might provide good approximations also in this case, for

weak couplings and low temperatures such that there's few excitations in the environment that effectively behave like bosons.

## 2.3 Numerical implementation and cost

With respect to standard master equations based on the Born approximation (1), equation (4) for the dynamical map is a non-linear integro-differential equation. Non-linear equations can in principle lead to numerical instabilities, as for example one might encounter discretizing Eq. (4) with the simplest, explicit Euler scheme. On the other hand, Eq. (4) has the form of a Dyson equation with a self-consistent self-energy, which is often encountered in many-body physics and for which several stable discretization schemes are known (see e.g. [110] and references therein). Here we use a simple and efficient implicit second-order Runge-Kutta scheme adapted from Ref. [110] and described in Appendix A, which is found to be numerically very stable (also in the case of a larger Hilbert space [103]).

Compared with Born-approximation-based master equations, the NCA maps can capture non-perturbative effects in the system-bath coupling, with a similar formalism and with a moderate increase in numerical cost. The NCA map has a computational cost for time-propagation which is $O(t^2)$, like the Born master equation, and the NCA-Markov map has cost $O(t)$, as Redfield [2] or Lindblad equations [101], allowing long propagation times. In the case of OCA (C.20) the cost gets an additional $O(t^2)$ factor, owing to the additional time-integrals involved. Note also that in the case of non-stationary environments or time-dependent Hamiltonians, there's an additional $O(t)$ cost, as the maps depend on two times rather than only on time differences. The main computational disadvantage with respect to master equations is dealing with $\hat{\mathcal{V}}$ which has size $N^4$, where $N$ is the size of the system Hilbert space, instead of with $\rho$ having size $N^2$. This is not an issue for systems with a small Hilbert space, while for larger systems the additional cost might be limiting, but is still moderate compared to exact methods for open quantum systems. In the latter case of large systems, the maps can be used in combination with other many-body approaches such as the Dynamical Mean-Field Theory [103,110,111] or could be compressed using memory-efficient methods such as matrix product operators [112–114].

# 3 Application to the spin-boson model

In the following, we demonstrate the NCA and NCA-Markov dynamical maps on the spin-boson model and compare them to the Born and Born-Markov master equations.

This model describes a quantum spin 1/2 with tunneling strength $\Delta$ coupled to a bosonic bath with Hamiltonian

$$H = \frac{\Delta}{2}\sigma^x + \frac{\lambda}{2}\sigma^z \sum_i \left(a_i^\dagger + a_i\right) + \sum_i \omega_i a_i^\dagger a_i. \tag{9}$$

Despite its simple Hamiltonian, the spin boson model has a non-trivial physics that has been previously investigated with many theoretical methods [70,115–134]. It can also be nowadays realized in several experimental setups [24,56,135]. Identifying $X = \sigma^z$ and $B = \frac{\lambda}{2}\sum_i \left(a_i^\dagger + a_i\right)$, then the bath correlation function $\Gamma(t)$ (3), is fixed by specifying the density of states of the bath modes and its temperature. We consider here a zero temperature bath with bath correlation function

$$\Gamma(\tau) = \int_0^\infty \frac{d\xi}{2\pi} J(\xi) \frac{1}{2} \left(\cos(\xi t) - i\sin(\xi t)\right), \tag{10}$$

and the bath density of states is taken of the form
$J(\omega) = \sum_i \pi\lambda^2 \delta(\omega - \omega_i) = 2\pi\alpha\omega_c^{1-s}\omega^s\theta(\omega)\theta(\omega_c - \omega)$, with a sharp cutoff $\omega_c$ and where

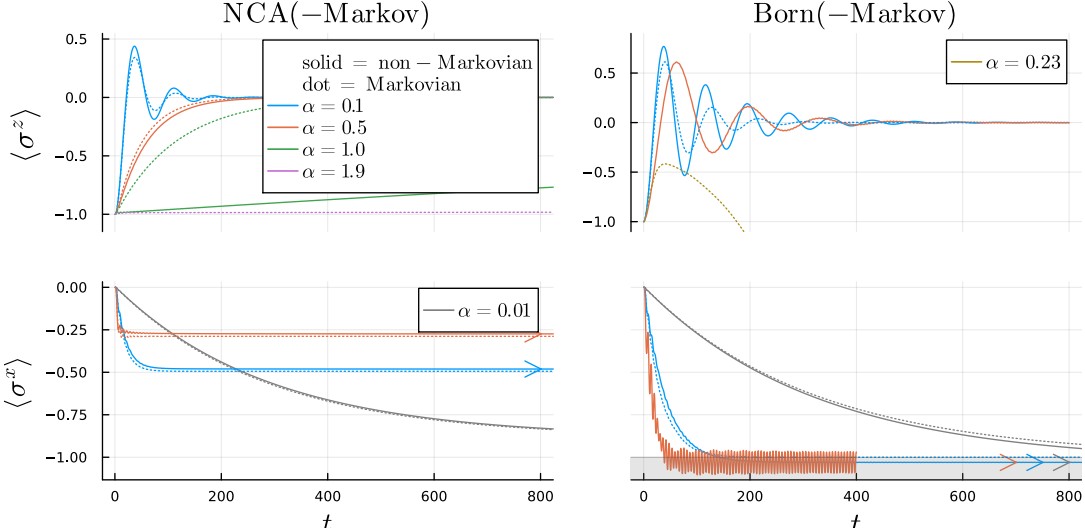

Figure 2: Dynamics of the spin boson model, for $\Delta/\omega_c = 0.1$, $\omega_c = 1$, computed with the NCA and NCA-Markov maps (left) and with the Born and Born-Markov master equations (right) for different values of the system-bath coupling $\alpha$, starting from $\rho(0) = |\downarrow\rangle\langle\downarrow|$. The top panels show the dynamics of $\langle\sigma^z\rangle$: on the left, NCA(-Markov) captures that the dynamics becomes incoherent for $\alpha > \alpha_{\text{incoh}}$; also, it captures that the timescale for relaxation diverges as $\alpha$ approaches a critical value $\alpha_c$, signaling the onset of the spin-boson delocalization transition. These features are not captured instead by the Born(-Markov) master equation (top-right).

The bottom panels show the relaxation of $\langle\sigma^x\rangle$ to its stationary value (indicated with arrows). NCA(-Markov) predicts a non-trivial dependence of the steady-state on the system-bath coupling, which is missing in Born and Born-Markov, even at small coupling. The right panels show the breakdown of the Born approximation, developing unphysical oscillations (bottom), and of the Born-Markov one, developing an unphysical instability (top). Parameters: $dt = 0.1 \times 2\pi/\omega_c$, apart for the Born dynamics where $dt = 0.01 \times 2\pi/\omega_c$.

$\alpha$ is the dimensionless system-bath coupling strength $\alpha = \lambda^2/(2\omega_c^2)$ (we fix $\omega_c = 1$ and $\Delta/\omega_c = 0.1$ throughout the manuscript). For $s = 1$ this corresponds to a Ohmic environment and for $s < 1$ to a sub-Ohmic one, that will be considered in the rest of the manuscript.

## 3.1 Ohmic spin-boson: NCA maps vs master equations

We consider here the Ohmic spin boson ($s = 1$) in the limit $\Delta \ll \omega_c$, that is well understood already from the seminal work [118]: For $\alpha < 1/2$, there are coherent oscillations (tunneling) between the $\sigma^z$ eigenstates at a renormalized spin frequency $\Delta_r$ which are damped by the environment, for $\alpha > 1/2$ then $\Delta_r$ is still finite but oscillations are overdamped and the dynamics is incoherent, while for $\alpha > \alpha_c = 1$ tunneling is suppressed and the spin enters a localized phase with $\Delta_r = 0$, through a quantum-phase transition.

Fig. 2 shows the transient dynamics of the system starting from $\rho(0) = |\downarrow\rangle\langle\downarrow|$, with $\sigma^z|\downarrow\rangle = -1|\downarrow\rangle$, with left and right columns corresponding to NCA maps and Born-based master equations. The spin-boson physics discussed above is captured both by the NCA and NCA-Markov maps, as shown in the top-left panel, plotting the dynamics of $\langle\sigma_z\rangle$. They capture a crossover between a coherent spin dynamics for $\alpha < \alpha_{\text{incoh}}$, characterized by underdamped oscillations, and an incoherent dynamics for $\alpha > \alpha_{\text{incoh}}$ where oscillations are overdamped. Numerically we locate the crossover at $\alpha_{\text{incoh}} \sim 0.2$, with a quantitative discrepancy with respect

to more accurate estimates [118]. We also note that in this regime the NCA and NCA-Markov agree remarkably well and that memory effects do not seem to play a major role. Increasing the system-bath coupling, we see that both NCA and NCA-Markov approaches predict a growth of the spin relaxation timescale as $\alpha$ approaches the critical value, which witnesses the onset of the localization quantum phase transition [121]. While the qualitative behaviour of the two theories is similar, the NCA approach predicts $\alpha_c \approx 1$, as expected [118], while NCA-Markov predicts a larger value $\alpha_c \approx 1.9$. The renormalization of the spin frequency down to zero at the localization transition can be equally inferred from the stationary-state correlation functions that we report in Appendix E.1. Remarkably, the NCA and NCA-Markov maps can qualitatively capture the non-perturbative physics of the spin-boson model.

On the other hand, the Born-based master equations fail to reproduce, even qualitatively, the crossover to an incoherent regime and the localization transition characterizing the spin boson model (see top-right panel). The Born-Markov approximation is numerically divergent already at small coupling (top-right panel), while the Born one, despite not diverging, never predicts a crossover to over-damped oscillations (bottom-right panel). That the Born-based master equations cannot predict the localization transition is also evident from the analysis of steady-state correlation functions in Appendix E.1.

The lower panels show instead the relaxation of $\langle\sigma_x\rangle$ to its stationary value (indicated by arrows), for different values of $\alpha$. For a vanishingly small system-bath coupling the spin thermalizes to its ground state, where $\langle\sigma^x\rangle = -1$, while increasing $\alpha$ the coupling with the bath drives the spin towards the $z$ direction, and thus $\langle\sigma^x\rangle$ decreases. We remark that the NCA maps reproduce the expected behaviour in the whole delocalized phase (bottom-left panel). Instead, the Born and Born-Markov master equations cannot capture any steady-state dependence on $\alpha$ (see Appendix E.3), and thus they cannot predict the $\langle\sigma^x\rangle$ dynamics correctly (bottom-right panel). Similar pathologies are known to severely limit these equations, even at small couplings [136, 137]. We also note that the Born master equation develops unphysical $\langle\sigma_x\rangle$ oscillations (bottom-right panel), reminiscent of an unphysical gap in the equilibrium $\sigma_x$ correlation functions [132], and unphysical $\langle\sigma_x\rangle < -1$ steady-state values of this quantity. These correspond to a non-positive density matrix, that instead the NCA maps never showed a tendency to develop, in the entire delocalized phase. Finally, we remark that the NCA maps in the present form cannot capture the localized phase of the spin-boson model, in which the bath develops finite coherence and finite anomalous correlation functions, but they can be extended also to that case.

In addition, to further highlight the ability of the NCA map to go beyond standard master equations, in Appendix E.1 we computed the steady-state correlation functions showing how the spin transition frequency is renormalized to zero at the localization transition in NCA, while this never happens for Born-based master equations. In Appendix E.2 we studied the response of the spin-boson model with a finite bias and shown that it reproduces the crossover to an incoherent regime observed experimentally in [135] that Born-based master equations fail at predicting.

## 3.2 A quantitative benchmark

Here we show that the NCA maps can give results that are quantitatively accurate beyond the capability of Born-approximation-based master equations. We consider a spin-boson model with both Ohmic and sub-Ohmic ($s < 1$) environments. In the latter case in fact, the localization transition happens at weaker system-bath couplings [132], influencing more the weak coupling regime where the NCA becomes quantitatively accurate.

We compute the leading-order OCA corrections, that are compared in Fig. 3 against the results of the NCA maps and of the Born-based master equations. The left panels show the real-time dynamics, for which it can be seen that the OCA corrections to NCA are negligible at

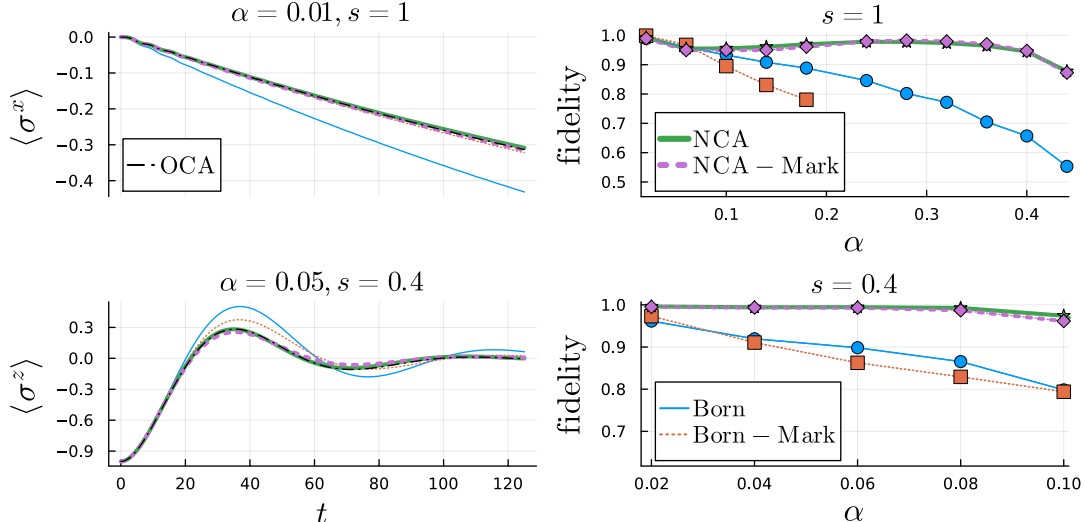

Figure 3: A comparison between NCA, OCA and the Born approximation, for a Ohmic (top panels) and sub-Ohmic $s = 0.4$ baths (bottom panels). The NCA predictions are shown to be quantitatively correct, since the OCA predictions are very close to the NCA ones. Respectively, the left panels show the real time dynamics, while left ones show the minimum process fidelity of the various maps compared to OCA as defined in the text. Note that in the upper-right plot, the Born-Markov points are plotted until $\alpha \approx 0.2$ as it diverges for larger couplings. The levels splitting and bath cutoff are $\Delta/\omega_c = 0.1, \omega_c = 1$.

all times, while the Born and Born-Markov equations show significant deviations. The top-left panel shows the dynamics of $\langle \sigma_x \rangle$ for a Ohmic environment, for which we already remarked (in Fig. 2) that the NCA maps capture a non-trivial steady-steady value, that the Born-based equations cannot predict. Accordingly, the latter equations display a large discrepancy also in the transient dynamics, while NCA results are accurate, as the OCA corrections are small. A similar conclusion is drawn from the bottom panel, where we show the dynamics of $\langle \sigma_z \rangle$ for a sub-Ohmic environment with $s = 0.4$. In this case, the zero steady-state value is correctly captured by all methods, yet the master equations show significant errors in the transient dynamics.

In the right panels instead we report an initial-state and observable-independent measure of the fidelity between the most accurate OCA map, and the other methods. As such a measure, we consider the Jamiolkowski process fidelity $F(\rho, \sigma) \equiv \text{tr}(\sqrt{\sqrt{\rho} \sigma \sqrt{\rho}})^2$ of the density-matrices associated with the respective maps, that indeed describe quantum processes, by the Jamiolkowski isomorphism [138]: $\rho_\mathcal{V} \equiv [\mathcal{I} \otimes \mathcal{V}](|\Phi\rangle\langle\Phi|)$, where $|\Phi\rangle = \Sigma_j |j\rangle|j\rangle/\sqrt{d}$ is a maximally entangled state of the ($d$-dimensional) system with another copy of itself, $\{|j\rangle\}$ is an orthonormal basis set and $\mathcal{I}$ the identity matrix. In the right panels of Fig. 3 we show the fidelity with OCA, defined as the minimum fidelity in the time-interval identified by the time-axis of the left panels (a measure of worst error). For both the Ohmic and sub-Ohmic case the NCA maps have better values of fidelity, up to much higher values of coupling: For the Ohmic case the fedelity starts to deteriorate around $\alpha \approx 0.5$, after which the NCA (and OCA) results are still qualitatively but not quantitatively correct, while in the sub-Ohmic case the fidelity is good until $\alpha \approx 0.1$, that is close to the localization transition critical point [132].

We finally remark that the NCA-Markov approach in Fig. 3 is in good agreement with NCA and OCA predictions, showing that the Born approximation is much more limiting than the Markovian approximation here.

## 4 Conclusion

In this work we have introduced the NCA and NCA-Markov maps dynamical maps for open quantum systems. These approaches are formally very similar to conventional master equations based on the Born and Born-Markov approximation and become equivalent to them in the weak-coupling limit, but can also capture non-perturbative effects and stronger couplings with a moderate increase in numerical cost and therefore might be preferred in several numerical studies of open quantum systems. We have also discussed how the NCA maps can be benchmarked evaluating the leading order self-consistent correction, the one-crossing approximation (OCA).

We applied the NCA maps to the spin-boson model at zero temperature, showing how they qualitatively capture its non-perturbative physics in presence of a Ohmic bath, including its crossover between coherent and incoherent dynamics and its phase transition towards a localized phase, signalled by a growth of the spin relaxation time-scale and a renormalization of the spin frequency to zero, which conventional weak-coupling master equations fail at predicting. We have also evaluated the leading-order OCA correction both for a Ohmic and a sub-Ohmic spin-boson model, showing that the NCA maps also yield quantitatively correct predictions in regimes in which standard master equations show significant deviations.

We expect the NCA maps to find a variety of applications beyond the model considered in this paper, including electronic transport problems [39–41], driven problems [139–141], non-perturbative environments such as nanophotonic structures [142–144] and collective effects of multiple emitters coupled to a common environment [145].

In the case of systems with a large Hilbert space, these maps can be combined with a Dynamical Mean-Field Theory approach [103,110,111], or could be compressed using memory-efficient methods such as matrix product operators [112–114].

## Acknowledgements

**Funding information** We acknowledge computational resources on the Collége de France IPH cluster. This work was supported by the ANR grant "NonEQuMat" (ANR-19-CE47-0001). It was also supported by the Engineering and Physical Sciences Research Council (EPSRC) and by the Science and Technology Facilities Council (STFC) [grant number EP/W005484].

**Data availability** The data to reproduce the results of the manuscript will be provided by the author upon request.

## A Implementation of a 2nd order Runge-Kutta scheme

Equation (4) is a Volterra integral-differential equation, that often appears in many-body problems, with the form

$$\frac{d}{dt}y(t) = q(t) + p(t)y(t) + \int_0^t d\bar{t}\, k(t,\bar{t})y(\bar{t}), \tag{A.1}$$

where in our case of an equilibrium bath the memory kernel depends only on time differences. Several stable discretization schemes can be used to solve it and here we implemented a 2nd order Runge-Kutta scheme adapted from [110], Appendix A, to propagate it in time.

In the present case we adapt that scheme, such as to capture correctly the case in which the memory kernel has a $\delta(t)$ contribution, for example in the case of a Markovian environment.

By integrating (A.1) up to the time-steps $t_n$ and $t_{n-1}$ and subtracting the two equations, we get

$$y(t_n) - y(t_{n-1}) = \Delta t \sum_{i=0}^{n-1} (w_{n,i} - w_{n-1,i}) y'(t_i) + \Delta t w_{n,n} y'(t_n), \tag{A.2}$$

which is Eq. A5 of [110], with $w_{n,i}$ $(i = (0, 1, \ldots, n))$ some weights that depend on the scheme chosen for approximating integrals (Euler, trapezoids, Simpson's rule, ecc). $y'(t_n)$ is then evaluated by discretizing (A.1). In doing so, we allow $k(t, \bar{t})$ to have a $\delta(t - \bar{t})$ contribution. To correctly take it into account, we discretize the integral as

$$\int_0^{t_n} d\bar{t}\, k(t_n, \bar{t}) y(\bar{t}) = \int_0^{t_{n-1}} d\bar{t}\, k(t_n, \bar{t}) y(\bar{t}) + \int_{t_{n-1}}^{t_n} d\bar{t}\, k(t_n, \bar{t}) y(\bar{t})$$
$$= \Delta t \sum_{i=0}^{n-1} w_{n-1,i} k(t_n, t_i) y(t_i) + \Delta t\, k(t_n, t_n) y(t_n), \tag{A.3}$$

where we discretized the last time-step separately, to make sure that $k(t_n, t_n)$ is evaluated with weight 1, such as to fully capture a $\delta(t - \bar{t})$ contribution (instead of discretizing the whole integral with a single scheme).

In our implementation, we use a second-order Runge-Kutta scheme for the weights $w_{n,i}$, corresponding to

$$w_{n,i} = \begin{cases} 1/2, & i = 0, n, \\ 1, & 1 \leq i \leq n-1, \end{cases} \tag{A.4}$$

leading to

$$y'(t_n) = q(t_n) + p(t_n) y(t_n) + \Delta t\, k(t_n, t_n) y(t_n)$$
$$+ \frac{\Delta t}{2} \left[ k(t_n, t_0) y(t_0) + k(t_n, t_{n-1}) y(t_{n-1}) \right] + \Delta t \sum_{i=1}^{n-2} k(t_n, t_i) y(t_i), \tag{A.5}$$

$$y(t_n) - y(t_{n-1}) = \frac{\Delta t}{2} \left[ y'(t_{n-1}) + y'(t_n) \right]. \tag{A.6}$$

Using those equations it is possibile to determine $y(t_n), y'(t_n)$, given their values for $t_{n'} < t_n$ and an initial condition for $y(t_0), y'(t_0)$ (in our case $\hat{\mathcal{V}}(0) = \mathbb{1}$ and $\partial_t \hat{\mathcal{V}}(0)$ is given by Eq. (4)). If $k(t, \bar{t})$ doesn't depend on $y$ one might eliminate $y'(t_n)$ and find a closed equation for $y(t_n)$. Since in our case $k(t, \bar{t})$ depends on $y$ as well, we iterate the 2 equations above to find $y(t_n), y'(t_n)$.

# B  System-bath coupling in the rotating wave approximation

Here we consider the case of a system-bath coupling in the form $\tilde{H}_{SB} = \tilde{X} \otimes \tilde{B}^\dagger + \tilde{X}^\dagger \otimes \tilde{B}$, which is common when performing a rotating-wave approximation. Here $\tilde{B} = \sum_i \frac{\tilde{\lambda}_i}{2} a_i$. We still consider a bosonic bath, while for fermionic ones we refer to [100]. In this case the NCA dissipator is given by

$$\hat{\tilde{\mathcal{D}}}(\tau) = \tilde{\Gamma}^<(\tau) \left( \tilde{X}^\dagger \hat{\mathcal{V}}(\tau)[\bullet] \tilde{X} - \hat{\mathcal{V}}(\tau)[\bullet \tilde{X}] \tilde{X}^\dagger \right)$$
$$+ \tilde{\Gamma}^>(\tau) \left( \hat{\mathcal{V}}(\tau)[\tilde{X} \bullet] \tilde{X}^\dagger - \tilde{X}^\dagger \hat{\mathcal{V}}(\tau)[\tilde{X} \bullet] \right) + \text{H.c.}, \tag{B.1}$$

with $\tilde{\Gamma}^>(\tau) = \text{tr}\left[ \tilde{B}(\tau) \tilde{B}^\dagger(0) \rho_B(0) \right]$ and $\tilde{\Gamma}^<(\tau) = \text{tr}\left[ \tilde{B}^\dagger(\tau) \tilde{B}(0) \rho_B(0) \right]$.

# C Formal derivation of the self-consistent dynamical maps

## C.1 Perturbation theory in the system-bath coupling

In this section we derive the perturbation series for the evolution superoperator in the system-bath coupling, up to all orders in this coupling. This leads to the Dyson equation (4) of the main text and sets the stage for developing the non-crossing approximation.

We recall that the evolution superoperator $\hat{\mathcal{V}}$ is defined by $\rho(t) = \mathrm{tr}_B \rho_{\mathrm{tot}}(t) = \hat{\mathcal{V}}(t)\rho(0)$ where $\rho_{\mathrm{tot}}(t) = e^{-iHt}\rho_{\mathrm{tot}}(0)e^{iHt}$ and the total Hamiltonian is given by $H = H_S + H_B + H_{SB}$ and where $\mathrm{tr}_B$ is a partial trace on bath operators. By moving to the interaction picture, the following identities can be found for the evolution operators

$$e^{-iHt} = e^{-i(H_S+H_B)t} T_t e^{-i \int_0^t dt' H_{SB}(t')} = e^{-i(H_S+H_B)t} T_t \sum_{n=0}^{\infty} \frac{(-i)^n}{n!} \left[ \int_0^t dt' H_{SB}(t') \right]^n, \quad \text{(C.1)}$$

$$e^{iHt} = \check{T}_t e^{i \int_0^t dt' H_{SB}(t')} e^{i(H_S+H_B)t} = \check{T}_t \sum_{n=0}^{\infty} \frac{(+i)^n}{n!} \left[ \int_0^t dt' H_{SB}(t') \right]^n e^{i(H_S+H_B)t}. \quad \text{(C.2)}$$

Here $H_{SB}(t') = e^{i(H_S+H_B)t'} H_{SB} e^{-i(H_S+H_B)t'}$ and $T_t$ is the real-time time-ordering operator. The latter takes any product of operators, where each operator is defined at one time, and changes the order so that every operator has only later operators to the left and earlier operators to the right. For non-fermionic variables as those we will consider, this change of order doesn't introduce any sign. The anti-time-ordering operator $\check{T}_t$ instead enforces the opposite ordering. Plugging in these expressions and using the cyclic property of the partial trace on the bath, one finds

$$\hat{\mathcal{V}}(t)\rho(0) = e^{-iH_St} \mathrm{tr}_B \left\{ T_t \sum_{k_+=0}^{\infty} \frac{(-i)^{k_+}}{k_+!} \left[ \int_0^t dt' H_{SB}(t') \right]^{k_+} \rho_{\mathrm{tot}}(0) \check{T}_t \sum_{k_-=0}^{\infty} \frac{(+i)^{k_-}}{k_-!} \left[ \int_0^t dt' H_{SB}(t') \right]^{k_-} \right\} e^{iH_St}. \quad \text{(C.3)}$$

In order to manage this double series expansion, it is convenient to use the superoperators notation already introduced in the main text. We define

$$\hat{H}_{SB\gamma}(t')\bullet = \begin{cases} H_{SB}(t')\bullet, & \text{if } \gamma = +, \\ \bullet H_{SB}(t'), & \text{if } \gamma = -. \end{cases} \quad \text{(C.4)}$$

Using this definition, we note that the time-ordering structure of the operators in Eq. (C.3), that are time-ordered on the left of the density matrix and anti-time-ordered on its right, is obtained in the superoperator notation if superoperators with $\gamma = +$ and $\gamma = -$ are time-ordered separately, putting superoperators with later times on the left regardless of their index $\gamma$ (instead of those with $\gamma = -$ being anti-time-ordered). We therefore introduce the operator $T_C$ enforcing this time-ordering of a string of superoperators (the notation recalls that of the contour-time-ordering operator in Keldysh field theory, which is analogous). This is easily understood with an example. Suppose $t_1 > t_2 > t_3 > t_4$, then

$$T_C \hat{O}_-(t_3)\hat{O}_-(t_4)\hat{O}_+(t_1)\hat{O}_+(t_2)\bullet = \hat{O}_-(t_3)\hat{O}_-(t_4)\hat{O}_+(t_1)\hat{O}_+(t_2)\bullet$$
$$= O(t_1)O(t_2) \bullet O(t_4)O(t_3) = T_t O(t_1)O(t_2) \bullet \check{T}_t O(t_3)O(t_4). \quad \text{(C.5)}$$

We also note that superoperators with different $\gamma$ indexes commute by definition, $\hat{O}_+(t_1)\hat{O}_-(t_2)\bullet = \hat{O}_-(t_2)\hat{O}_+(t_1)\bullet = O(t_1) \bullet O(t_2)$, thus we can always assume that a string of superoperators under $T_C$ is time-ordered according to their real-time variable, regardless of their indexes $\gamma$. Using the previous example

$$T_C \hat{O}_-(t_3)\hat{O}_-(t_4)\hat{O}_+(t_1)\hat{O}_+(t_2) = \hat{O}_-(t_3)\hat{O}_-(t_4)\hat{O}_+(t_1)\hat{O}_+(t_2)$$
$$= \hat{O}_+(t_1)\hat{O}_+(t_2)\hat{O}_-(t_3)\hat{O}_-(t_4), \quad \text{(C.6)}$$

where in the last equality the superoperators are ordered according to their real time variables $t_1 > t_2 > t_3 > t_4$. We also define the bare evolution superoperator $\hat{\mathcal{V}}_0(t) = e^{-iH_S t} \bullet e^{iH_S t}$. With these definitions one can write

$$\hat{\mathcal{V}}(t)\rho(0) = \hat{\mathcal{V}}_0(t)\mathrm{tr}_B\left\{T_C \sum_{k_+,k_-=0}^{\infty} \frac{(-i)^{k_+ + k_-}}{k_+!k_-!}\left[\int_0^t dt' \hat{H}_{SB+}(t')\right]^{k_+}\left[-\int_0^t dt' \hat{H}_{SB-}(t')\right]^{k_-}\rho_{\mathrm{tot}}(0)\right\}.$$
(C.7)

By doing some combinatorial calculations, it's easy to show that the double summation can be reduced to a single one and that the following identity is valid:

$$\hat{\mathcal{V}}(t)\rho(0) = \hat{\mathcal{V}}_0(t)\mathrm{tr}_B\left\{T_C e^{-i\int_0^t dt' \sum_{\gamma \in \{+,-\}} \gamma \hat{H}_{SB\gamma}(t')}\rho_{\mathrm{tot}}(0)\right\}$$
(C.8)

$$= \hat{\mathcal{V}}_0(t)\mathrm{tr}_B\left\{T_C \sum_{k=0}^{\infty} \frac{(-i)^k}{k!}\left[\int_0^t dt' \sum_{\gamma \in \{+,-\}} \gamma \hat{H}_{SB\gamma}(t')\right]^k \rho_{\mathrm{tot}}(0)\right\}.$$
(C.9)

We assume that there are initially no correlations between the system and the bath, that is $\rho_{\mathrm{tot}}(0) = \rho(0) \otimes \rho_B$, such that we can perform the partial trace over bath operators stemming from $H_{SB}(t') = X(t') \otimes B(t')$, raised to the $k$-th power. We take the bath state to be stationary under the bare bath Hamiltonian, for example to be in equilibrium at some bath temperature, and, without of loss of generality, we assume that $\mathrm{tr}_B(B(t)\rho_B(0)) = \mathrm{tr}_B(B\rho_B(0)) = 0$: a finite value of $\mathrm{tr}_B(B\rho_B(0))$ could be in fact always absorbed in the definition of the Hamiltonian [30]. Under this assumption, only terms with even powers of $k$ are non-zero, and therefore we change the summation index to sum only on those even terms. Finally we consider system and bath operators that are non-fermionic, such that they commute under time-ordering. Under these assumptions, we obtain

$$\hat{\mathcal{V}}(t)\rho(0) = \hat{\mathcal{V}}_0(t)\sum_{k=0}^{\infty}\frac{(-i)^{2k}}{(2k)!}\int_0^t dt_1 \cdots \int_0^t dt_{2k}$$
$$\times \sum_{\gamma_1 \cdots \gamma_{2k}} \gamma_1 \dots \gamma_{2k} \mathrm{tr}_B\left[T_C \hat{B}_{\gamma_1}(t_1)\dots\hat{B}_{\gamma_{2k}}(t_{2k})\rho_B(0)\right] T_C \hat{X}_{\gamma_1}(t_1)\dots\hat{X}_{\gamma_{2k}}(t_{2k})\rho(0).$$
(C.10)

Since the bath Hamiltonian $H_B = \sum_i \omega_i a_i^\dagger a_i$ is quadratic in its creation and annihilation operators and assuming also its initial state $\rho_B(0)$ is, we can use the Wick's theorem to simplify the multi-point correlators of bath operators. For Hermitian bosonic operators such as $B = \sum_i \frac{\lambda_i}{2}\left(a_i + a_i^\dagger\right)$ one can use the Wick's theorem for real bosonic variables yielding

$$\mathrm{tr}_B\left[T_C \hat{B}_{\gamma_1}(t_1)\dots\hat{B}_{\gamma_{2k}}(t_{2k})\rho_B(0)\right] = \sum_{\substack{\text{pairings of} \\ \{(\gamma_1,t_1),\dots(\gamma_{2k},t_{2k})\}}} \Gamma_{\gamma_{i_1}\gamma_{i_2}}(t_{i_1} - t_{i_2})\dots\Gamma_{\gamma_{i_{2k-1}}\gamma_{i_{2k}}}(t_{i_{2k-1}} - t_{i_{2k}}).$$
(C.11)

On the left-hand side, the two-point correlation functions of the bath appear

$$\Gamma_{\gamma,\gamma'}(t - t') = \mathrm{tr}_B\left[T_C \hat{B}_\gamma(t)\hat{B}_{\gamma'}(t')\rho_B(0)\right],$$
(C.12)

which depend on time differences because we assumed $\rho_B(0)$ to be stationary, and the sum runs over all the possible ways of forming pairs with the indexes $\{(\gamma_1, t_1), \dots(\gamma_{2k}, t_{2k})\}$. Also, since the integrand is completely symmetric by permuting two integration/summation indexes $(t_i, \gamma_i)$ and $(t_j, \gamma_j)$, we can limit the integration to the domain defined by $t_1 > t_2 > \cdots > t_{2k}$ and multiply by $(2k)!$. We can also drop the contour time-ordering operator $T_C$, which is at this point useless since time-ordering is already enforced by the extremes of integration. Finally, we write the string of system operators by making explicit their time-evolution as follows

$$\hat{\mathcal{V}}_0(t)\hat{X}_{\gamma_1}(t_1)\dots\hat{X}_{\gamma_{2k}}(t_{2k}) = \hat{\mathcal{V}}_0(t - t_1)\hat{X}_{\gamma_1}\hat{\mathcal{V}}_0(t_1 - t_2)\dots\hat{X}_{\gamma_{2k}}\hat{\mathcal{V}}_0(t_{2k}),$$
(C.13)

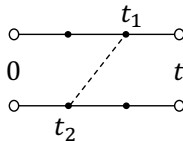

Figure 4: The diagrammatic representation of the term (C.15).

and we drop the initial state $\rho(0)$, since this is arbitrary. Eventually, we obtain the series for the evolution superoperator $\hat{\mathcal{V}}$:

$$
\hat{\mathcal{V}}(t) = \sum_{k=0}^{\infty} (-i)^{2k} \int_0^t dt_1 \int_0^{t_1} dt_2 \cdots \int_0^{t_{2k-1}} dt_{2k} \sum_{\gamma_1 \cdots \gamma_{2k}} \gamma_1 \ldots \gamma_{2k} \sum_{\substack{\text{pairings of} \\ \{(\gamma_1,t_1),\ldots(\gamma_{2k},t_{2k})\}}}
$$
$$
\times \left[ \Gamma_{\gamma_{i_1}\gamma_{i_2}}(t_{i_1} - t_{i_2}) \ldots \Gamma_{\gamma_{i_{2k-1}}\gamma_{i_{2k}}}(t_{i_{2k-1}} - t_{i_{2k}}) \right] \hat{\mathcal{V}}_0(t-t_1)\hat{X}_{\gamma_1} \hat{\mathcal{V}}_0(t_1-t_2)\hat{X}_{\gamma_2} \ldots \hat{X}_{\gamma_{2k}} \hat{\mathcal{V}}_0(t_{2k}).
$$
$$(C.14)$$

While it is possible to manipulate this series directly, it is more convenient for our purposes to show that it takes the form of the Dyson equation (4) of the main text and to introduce its self-energy. This is the object of the next section.

## C.2  Feynman diagrams, self-energy and Dyson equation

In order to show that Eq. (C.14) can be cast in the Dyson form of Eq. (4) of the main text, we start by introducing its diagrammatic representation. Each term of the series with fixed $k, t_1 \ldots t_{2k}, \gamma_1 \ldots \gamma_{2k}$ and with a fixed choice of pairings can be represented as a Feynman diagram following these rules:

- draw two parallel solid lines, each representing a portion of time axis going from time $t = 0$ to time $t$.

- locate the $t_1 > \cdots > t_{2k}$ times on those axes, where the $k_+$ times corresponding to an $\hat{X}_+$ superoperator must be drawn on the first time axis and the $k_- = k - k_+$ times corresponding to an $\hat{X}_-$ superoperator on the other one. The couple of solid-line segments going from $t_i$ to $t_{i+1}$ represent the evolution superoperator $\hat{\mathcal{V}}_0(t_{i+1} - t_i)$.

- connect the times paired by $\Gamma$ functions with dashed lines.

Finally, one also needs to keep track of the factor $(-1)^{2k}$ as well as of the sign given by the product $\gamma_1 \ldots \gamma_{2k}$. As an example, the term with expression

$$
\Gamma_{+-}(t_1 - t_2)\hat{\mathcal{V}}_0(t - t_1)\hat{X}_+ \hat{\mathcal{V}}_0(t_1 - t_2)\hat{X}_- \hat{\mathcal{V}}_0(t_2)\bullet, \tag{C.15}
$$

corresponds to the diagram in Fig. 4: Summing over the $\gamma$ indexes, one can also define more compact diagrams where the double-time axis is collapsed on a single time-axis, as shown in Fig. 5: From these diagrammatic rules, we can define a "self-energy" of the series on the lines of textbook calculations in many-body theory. For this purpose, we define as one-particle-irreducible (1PI) the compact diagrams which cannot be separated, by cutting a solid line, in two parts that are not connected by any dashed line. An example of 1PI diagrams is given in Fig. 6. Then, the self-energy $\hat{\mathcal{D}}$ is defined as the sum of all 1PI diagrams with the first an last solid lines removed: its diagrammatic representation is given in Fig. 6.

The superoperator $\hat{\mathcal{V}}$ is given by the sum of all diagrams, both 1PI and non-1PI. The definition of the self-energy is useful because all non-1PI diagrams can be obtained by joining some

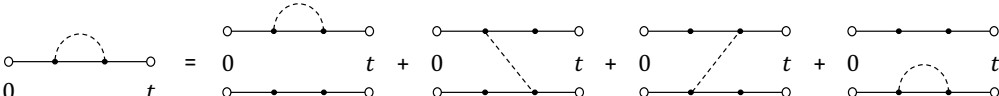

Figure 5: Single time-axis diagrams represent the set of diagrams where super-operator indexes are summed over.

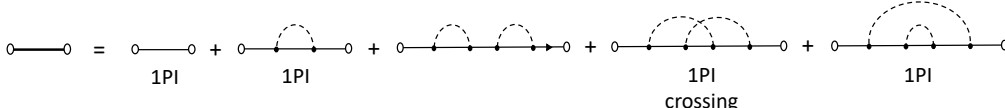

Figure 6: 1-particle-irriducible (1PI) diagrams making up the self-energy and crossing diagrams.

1PI diagrams with solid lines and, therefore, the whole series for $\hat{\mathcal{V}}$ can be written as

$$\hat{\mathcal{V}} = \hat{\mathcal{V}}_0 + \hat{\mathcal{V}}_0 \circ \hat{\mathcal{D}} \circ \hat{\mathcal{V}}_0 + \hat{\mathcal{V}}_0 \circ \hat{\mathcal{D}} \circ \hat{\mathcal{V}}_0 \circ \hat{\mathcal{D}} \circ \hat{\mathcal{V}}_0 + \dots,$$

where the circle operator "$\circ$" stands for partial time convolutions as in Eq. (C.14). This series sums up to the Dyson equation

$$
\begin{aligned}
\hat{\mathcal{V}}(t) &= \hat{\mathcal{V}}_0(t) + \int_0^t dt_1 \int_0^{t_1} dt_2 \hat{\mathcal{V}}_0(t - t_1) \hat{\mathcal{D}}(t_1 - t_2) \hat{\mathcal{V}}(t_2) \\
&= \hat{\mathcal{V}}_0(t) + \int_0^t dt_1 \int_0^{t_1} dt_2 \hat{\mathcal{V}}(t - t_1) \hat{\mathcal{D}}(t_1 - t_2) \hat{\mathcal{V}}_0(t_2),
\end{aligned}
\tag{C.16}
$$

or equivalently, to the integro-differential form reported as Eq. (4) in the main text:

$$\partial_t \hat{\mathcal{V}}(t) = \hat{\mathcal{H}}_S \hat{\mathcal{V}}(t) + \int_0^t dt_1 \hat{\mathcal{D}}(t - t_1) \hat{\mathcal{V}}(t_1). \tag{C.17}$$

### C.3 The non-crossing approximation

The non-crossing approximation (NCA) corresponds to approximating the series for $\hat{\mathcal{V}}$, and thus also for $\hat{\mathcal{D}}$, by keeping only the compact diagrams in which dashed lines do not cross. We remark that considering the compact diagrams is important for defining non-crossing diagrams, as in their non-compact version ambiguities arise as dashed lines might cross or not depending on how one draws them. An example of crossing diagram is given in Fig. 6 and the diagrammatic representation of the NCA self-energy is shown in Fig. 1.

It turns out that the NCA self-energy coincides with the $k = 1$ term of the exact self-energy, where the bare propagator $\hat{\mathcal{V}}_0$ is replaced with the dressed one $\hat{\mathcal{V}}$. This statement corresponds to the second equality in Fig. 1. In fact, we remark that the first and last times of a self-energy diagram must be connected together by a dashed line in the non-crossing approximation: if it's not the case, in fact, the resulting diagram is either non-1PI or it's crossed. We also remark that all diagrams with first and last time connected are necessarily 1PI. Therefore all the diagrams of $\hat{\mathcal{D}}_{\text{NCA}}$ are obtained connecting the remaining times in all possible non-crossing ways, but the latter diagrams in turn sum up to $\hat{\mathcal{V}}$, proving the equality.

The explicit expression of the NCA self-energy is then given by

$$\hat{\mathcal{D}}_{\text{NCA}}(\tau) = (-i)^2 \sum_{\gamma_1 \gamma_2} \gamma_1 \gamma_2 \Gamma_{\gamma_1 \gamma_2}(\tau) \hat{X}_{\gamma_1} \hat{\mathcal{V}}(\tau) \hat{X}_{\gamma_2}. \tag{C.18}$$

Carrying out the sum over $\gamma$ indexes and after some algebraic manipulations to go from the time-ordered correlation function $\Gamma_{\gamma_1,\gamma_2}(\tau)$, defined in Eq. (C.12), to the non-time ordered correlation function $\Gamma(\tau)$, defined in Eq. (3) of the main text, the NCA self-energy reduces to Eq. (5) of the main text.

### C.4  Validity of the NCA and Markov approximations together

We discuss in this appendix the validity of the NCA-Markov approximation, leading to Eq. (6) of the main text. At first glance the assumptions behind the NCA and Markovian approximations seem to contradict each other. The Markovian approximation assumes that the bath-induced dynamics on the timescale of bath correlation functions, call it $\tau_b$, is negligible, thus one can approximate $\hat{\mathcal{V}}(t-\tau)$ with $e^{-\hat{\mathcal{H}}_s\tau}\hat{\mathcal{V}}(t)$. The NCA self-energy, one the other hand, captures bath-induced processes happening on timescales shorter than $\tau_b$. This contradiction is resolved and the two approximations can be made simultaneously, if one assumes that bath-induced dynamics, on a timescale $\tau_b$, can be neglected only when $t > \tau_b$. In this way the Markovian approximation on the NCA master-equation is legitimate:

$$
\begin{aligned}
\partial_t \hat{\mathcal{V}}(t) &= \hat{\mathcal{H}}_S \hat{\mathcal{V}}(t) + \int_0^t d\tau \left[ \Gamma(\tau)(-X_+ + X_-)\hat{\mathcal{V}}(\tau)X_+ + \text{H.c.} \right] \hat{\mathcal{V}}(t-\tau) \\
&\approx \hat{\mathcal{H}}_S \hat{\mathcal{V}}(t) + \int_0^t d\tau \left[ \Gamma(\tau)(-X_+ + X_-)\hat{\mathcal{V}}(\tau)X_+ + \text{H.c.} \right] e^{-\hat{\mathcal{H}}_s\tau}\hat{\mathcal{V}}(t),
\end{aligned}
\tag{C.19}
$$

where $\hat{\mathcal{V}}(t) \approx e^{\hat{\mathcal{H}}_s\tau}\hat{\mathcal{V}}(t-\tau)$ for $t > \tau_b$ has been approximate in the spirit of a Markovian approximation, while $\hat{\mathcal{V}}(\tau)$ is probed inside the integral only at times $\tau < \tau_b$ and thus bath-induced dynamics is not neglected for this propagator. The essence of the NCA-Markov approximation, therefore, is that the bath-induced dynamics at short times $t < \tau_b$ is feedbacked into the dynamics at $t > \tau_b$. To be fully consistent, one should evolve $\hat{\mathcal{V}}(t)$ up to $t \sim \tau_b$ with the non-Markovian NCA equations and then continue the propagation using the NCA-Markov master equation. The results of the main text have been obtained instead by integrating the NCA-Markov equations from $t = 0$.

We remark that the dynamics on the timescale $\tau_b$ is left completely unresolved in most Markovian master equations [33,107,109], where the upper integration integral in Eq. (C.19) is sent to infinity.

### C.5  Beyond NCA

It is possible to go systematically beyond the NCA, which is particularly useful to assess the validity of its predictions. The exact self-energy $\hat{\mathcal{D}}$, which is represented in Fig. 1 as a series of diagrams in terms of the "bare" propagator $\hat{\mathcal{V}}_0 = e^{\hat{\mathcal{H}}_s\tau}$, can be also expressed as a series of "skeleton" (dressed) diagrams, which depend only on $\hat{\mathcal{V}}$ rather than on $\hat{\mathcal{V}}_0$. This is a standard result of many-body theory (see e.g. [104]). This skeleton series is defined by all the diagrams which have no self-energy insertions, that is no pieces that disconnect from the diagram by cutting two solid lines [104]. The NCA yields the first term of the skeleton series for the self-energy. We remark that each diagram of the skeleton series contains contributions up to all powers in the system-bath coupling. Nevertheless an "order" in the series exists such that higher-order diagrams are smaller than lower-order ones for a sufficiently weak coupling, as it is the case for the bare series. The natural strategy to assess the validity of the NCA results is therefore to include higher-order contributions to the self-energy. At sufficiently small coupling the NCA predictions become quantitatively accurate, as higher-order diagrams are negligible, while at stronger coupling one can still check qualitative agreement upon including higher-

order terms. Adding the leading-order diagram beyond NCA yields the self-energy

$$\hat{\mathcal{D}}_{\text{OCA}}(\tau) = \hat{\mathcal{D}}_{\text{NCA}}(\tau) + (-i)^4 \sum_{\gamma_1 \gamma_2 \gamma_0 \gamma} \gamma_1 \gamma_2 \gamma_0 \gamma$$

$$\times \int_0^\tau d\tau_1 \int_0^{\tau_1} d\tau_2 \Gamma_{\gamma\gamma_2}(\tau - \tau_2) \Gamma_{\gamma_1 \gamma_0}(\tau_1) \hat{X}_\gamma \hat{\mathcal{V}}(\tau - \tau_1) \hat{X}_{\gamma_1} \hat{\mathcal{V}}(\tau_1 - \tau_2) \hat{X}_{\gamma_2} \hat{\mathcal{V}}(\tau_2) \hat{X}_{\gamma_0}.$$

$$(\text{C.20})$$

This is known as one-crossing approximation (OCA) [94, 105], as it corresponds to summing the bare diagrams in which 2 dashed lines, corresponding to bath 2-times correlation functions, cross at most once. The second term of (C.20) becomes smaller than the first at a small enough coupling, because it is involves two bath correlation functions. As we mentioned in the main text, diagrams of the skeleton series are also ordered in terms of increasing decay-time of the self-energy or "memory" of the bath. In Fig. 1 we show that the NCA self-energy decays in a timescale $\tau \sim \tau_b$, in which the bath correlation function $\Gamma(\tau)$ decays: this is the case because $\hat{\mathcal{D}}_{\text{NCA}}(\tau)$ is proportional to $\Gamma(\tau)$ (Eq. (5) of the main text). The second term in Eq. (C.20) instead decays in $2\tau_b$, because of the two bath correlation functions ($\Gamma_{\gamma\gamma'}$ reduces to $\Gamma$ or to its complex conjugate given the time-ordering of its arguments).

An application of the OCA to the Spin-Boson model is reported in 3.2

# D  The Born-Markov master equation

A Markovian approximation of the Born master equation (1) leads to

$$\partial_t \rho(t) = \hat{\mathcal{H}}_S \rho(t) + \int_0^t d\tau \hat{\mathcal{D}}_{\text{Born}}(\tau) e^{-\hat{\mathcal{H}}_s \tau} \rho(t),$$

$$(\text{D.1})$$

which together with (2) is the Born-Markov master equation, used in the main text for comparison with the NCA-Markov map.

The most common way of writing this equation (see e.g. [31]) is by explicitly replacing the self-energy (2) in (D.1) getting

$$\partial_t \rho(t) = -i[H_s, \rho(t)] + \left[ X\tilde{X}\rho(t) + \tilde{X}\rho(t)X + \text{H.c.} \right],$$

$$(\text{D.2})$$

in terms of the "filtered" operators

$$\tilde{X} = \int_0^t d\tau \Gamma(\tau) e^{-iH_s \tau} X e^{iH_s \tau}.$$

$$(\text{D.3})$$

In the main text, we always keep the integration time in (D.3) finite to compare with the NCA-Markov map. Instead, when doing a fully Markovian approximation the integration in (D.3) is extended up to infinity [109], as bath correlation functions decay on a timescale $\tau \sim \tau_b$, while the system is assumed to change on a much slower timescale. This allows to evaluate the operator $\tilde{X}$ in the basis of the eigenstates of the system $|E_n\rangle$ with eigenvalues $E_n$, where it is expressed in terms of the Fourier transform of the retarded component of the bath correlation function $\Gamma^R(\omega) = \int_{-\infty}^\infty d\tau e^{i\omega\tau} \Gamma(\tau) \theta(\tau)$:

$$\tilde{X} = \sum_{nm} \Gamma^R(E_m - E_n)\langle n|X|m\rangle|n\rangle\langle m|.$$

$$(\text{D.4})$$

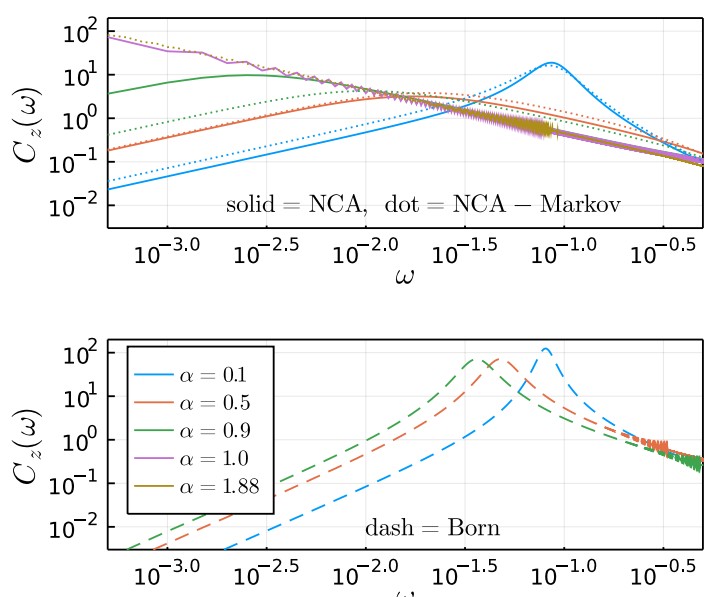

Figure 7: Steady-state correlation function of the $z$−spin component, $C_z(\omega)$ obtained through the NCA and NCA-Markov (top panel) quantum dynamical maps using Eq. (8), and their Born approximation (bottom panel). We see that both NCA approaches correctly capture the renormalization of the spin frequency $\Delta_r$ due to the bath coupling, ultimately leading to a quantum phase transition into a localized phase when $\Delta_r = 0$ for $\alpha = \alpha_c$. The low frequency behavior shows a power-law behavior of the form $C_z(\omega) \sim \omega$ as expected for an Ohmic bath. On the other hand the Born approximation fails both in the capturing the frequency renormalization and in the power-law regime at low frequency, different from the expected linear behavior.

# E  Further results on the spin-boson model

## E.1  Steady state correlation functions

In the main text, we have discussed the transient dynamics of the Ohmic spin-boson model, while here we focus on its correlation functions computed once the steady-state of the dynamics has been reached.

In Fig. 7 we show the Fourier transform of the steady-state correlation function $C_z(t) = \frac{1}{2}\lim_{t'\to\infty}\langle[\sigma_z(t+t'),\sigma_z(t')]\rangle$, which is a purely real function, computed from Eq. (8). At weak coupling with the bath, this correlation function shows a peak at $\Delta$, corresponding to spin-flip transitions at the frequency of the bare spin. Increasing the coupling to the bath, the spin frequency gets renormalized to a smaller value $\Delta_r$, which appears as a shift of the corresponding peak in $C_z(\omega)$. The upper panel of Fig. 7 shows that, increasing the system-bath coupling, both the NCA and NCA-Markov approaches predict such a renormalized spin frequency $\Delta_r$, which approaches $\Delta_r = 0$ as the critical coupling $\alpha_c$ is reached. This is known to happen [118,119] for the Ohmic spin-boson model at zero temperature for $\Delta/\omega_c \ll 1$. Note though that the exact functional dependence of $\Delta_r$ on $\alpha$ close to the critical point $\Delta_r = c\Delta(\Delta/\omega_c)^{\alpha/(1-\alpha)}$ [119,146] (where $c$ is a constant prefactor) is not correctly reproduced by the NCA. The Born-Markov approximation is not shown in Fig. 7 as it is nu-

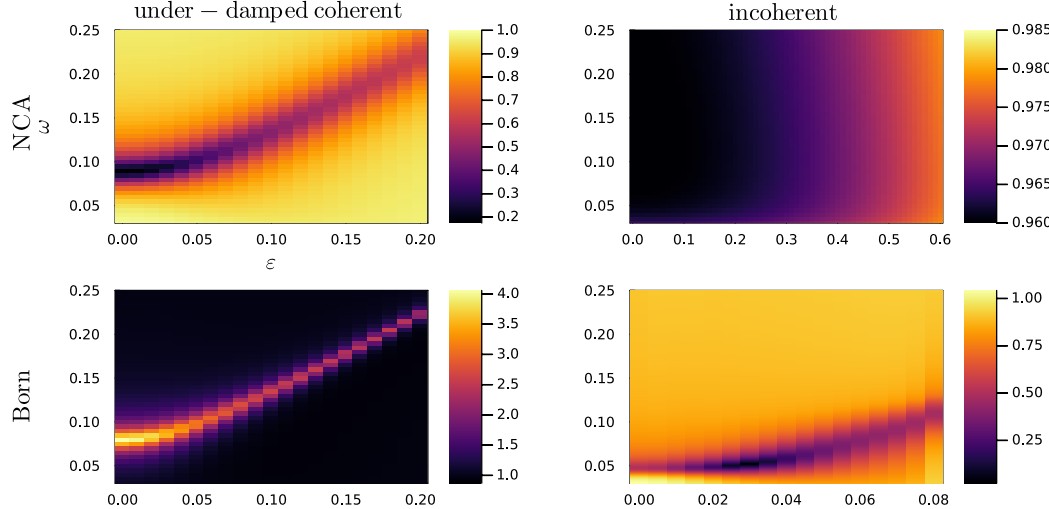

Figure 8: Modulus squared transmission $|\mathcal{T}(\omega)|^2$ of an applied probe field of frequency $\omega$ as a function of the two-level system bias $\epsilon$. The top panels are obtained using the NCA map, while the bottom ones with a Born master equation. Left(right) panels correspond to the regime of underdamped-coherent (incoherent) dynamics for $\alpha = 0.1(0.6)$. The NCA map captures the crossover between underdamped and incoherent dynamics, while the Born master equation always predict an underdamped dynamics.

merically unstable for $\alpha = 0.5, 0.9$ considered, while the Born approximation predicts only a very small shift of the peak, which never reaches zero for the values of coupling explored (up to $\alpha \approx 200$).

In addition, we also remark that the power-law behaviour of $C_z(\omega)$ at small $\omega$ is correctly captured in the NCA and NCA-Markov approximations, $C_z(\omega) \sim \omega$, while the Born approximation predicts $C_z(\omega) \sim \omega^2$.

### E.2 Connections to experiments

In a recent experiment [135], the spin-boson model is realized by a superconducting transmission line, whose electromagnetic modes represent the bosonic environment, coupled to a qubit, realizing the two-state system. The response of the system can be probed by applying a weak probe field at a frequency $\omega$ to the transmission line and measuring its transmission

$$\mathcal{T}(\omega) = 1 - i\mathcal{N}\omega\chi(\omega), \tag{E.1}$$

where $\chi(\omega)(t - t') = -i\langle[\sigma^z(t), \sigma^z(t')]\rangle\theta(t)$ is the retarded Green function of $\sigma^z$ and $\mathcal{N}$ is a coupling constant that we set to $\mathcal{N} = 1$, as we aim at a qualitative comparison with Ref. [135].

The crossover between underdamped-coherent and incoherent dynamics of the spin boson model can probed by transmission measurements scanning both the frequency of the probe $\omega$ and a static magnetic flux applied to the qubit $\epsilon$, resulting in the qubit Hamiltonian

$$H_{\text{qb}} = \frac{\Delta}{2}\sigma^x + \frac{\epsilon}{2}\sigma^z. \tag{E.2}$$

In Fig. 8 the color code indicates the modulus squared transmission $|\mathcal{T}(\omega)|^2$ as a function of $\omega$ and $\epsilon$ (in units of $\omega_c$). The first column refers to the regime of underdamped-coherent dynamics for $\alpha = 0.1$ while the second column to the incoherent dynamics regime for $\alpha = 0.6$;

the top panels are obtained using the NCA map, while the bottom ones with a Born master equation. The NCA results are in qualitative agreement with the experimental results of [135] (their Fig. 2), showing that the qubit dispersion relation can be read out of the transmission measurement in the underdamped-coherent regime (left panel), while in the incoherent regime (right panel) the transmission is nearly independent of the probe frequency $\omega$. In the bottom panels we show that the Born master equation does not capture the transition to a regime of incoherent dynamics and always predicts a trace of the qubit dispersion in the transmission (bottom right panel). This result agrees with the results on the real-time dynamics described in the main text (Fig. 2). We also remark that the Born master equation wrongly predicts enhanced, rather than suppressed, transmission for values of $\omega$ and $\epsilon$ hitting the qubit dispersion for $\alpha = 0.1$ (left plot).

### E.3   Steady-state dependence in NCA vs Born

The steady-state equation (7) also allows to understand why the NCA approaches capture the dependence of the steady-state on the system-bath coupling $\alpha$ in the top panels of Fig. 2 of the main text, which is instead missing in Born and Born-Markov theories: From our calculations we find that the steady-state of the spin-boson model does not depend on the Hamiltonian and takes the form $\rho_s = p_-|-\rangle\langle-| + p_+|+\rangle\langle+|$. In this case then Eq. (7) reduces to $\int_0^\infty d\tau \hat{\mathcal{D}}(\tau)\rho_s = 0$.

By expanding the self-energy $\hat{\mathcal{D}}(\tau)$ in powers of the system-bath coupling $\alpha$ as $\hat{\mathcal{D}}(\tau) = \alpha\hat{\mathcal{D}}^{(1)}(\tau) + \alpha^2\hat{\mathcal{D}}^{(2)}(\tau) + \ldots$, the Born and Born-Markov approximations correspond to truncating at first order in $\alpha$ and thus $\alpha$ drops in the steady-state equation: $\int_0^\infty d\tau \hat{\mathcal{D}}^{(1)}(\tau)\rho_s = 0$. Instead, the NCA and NCA-Markov approaches retain higher-order contributions to the self-energy, yielding a non-trivial dependence of the steady-state on $\alpha$.

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
