# Peer review of "Self-consistent dynamical maps for open quantum systems"

_SciPost Physics, doi:SciPost Phys. 16, 026 (2024)_

## Round 5 · Referee Report · Anonymous (Referee 1) · 2023-6-29

Strengths

  1. Original approach
  2. Detailed but clear manuscript

Weaknesses

  1. Qualitatively better results are demonstrated, but there are some obvious disagreements with respect to the well-known values, which somewhat undermine the claim that the method is applicable to the strong coupling regime.
  2. There are some minor loose ends in terms of the general applicability of the method.

Report

The authors present a scheme for computing the dynamical map of an open quantum system evolution, based on a perturbative diagrammatic approach inspired by many-body theory. The method is presented in a concise way, with the main text including an intuitive introduction to the main idea and the rigorous derivation in the appendix, where a numerical algorithm used to implement the scheme is also described.

Overall, the manuscript provides an interesting approach to tackling a well-known problem, with sufficient explanations, as well as enough detail for the reader to reproduce the results. However, after reading the manuscript, I believe that there are some open questions that need to be addressed before publication.

Requested changes

  1. In order to support the authors’ claim, beyond qualitative arguments, that the presented method can be used in the stronger coupling regime, I would suggest that a plot of the difference (perhaps in terms of the process fidelity or similar distance measure) between the NCA and OCA maps (as well as Born – OCA and Born-Markov - OCA) versus the coupling strength of the model is added. This would allow the reader to see: (a) Up to which coupling strength can we expect quantitatively reliable results, (b) how much of an improvement do we see compared to the standard approximations. It would be even more interesting to see a comparison to numerically exact results, however I believe that the OCA approximation can serve as a sufficient guideline for that, up to a certain coupling strength.
  2. Furthermore, while the authors claim that the method is general, in the rigorous derivation it is assumed that the bath operators commute under time ordering and later Wick’s theorem for real bosonic variables is applied. In order to claim generality it is necessary to comment on how these steps can be generalized to fermionic or spin environments. If this is not possible it should be mentioned in the main text.
  3. Some commentary on the feasibility of going to higher order approximations beyond NCA and OCA would be interesting to see.
  4. In the introduction, the authors mention that non-Markovian approaches are needed to describe the dynamics of superconducting circuits (SC). One of the three references [47-49] they have cited has nothing to do with SCs, while the other two are implementations of the spin-boson model with SC circuits. From this it is hard to argue that there is really a need to describe the dynamics of these systems beyond the Markovian regime. However, non-Markovian effects are also present in such systems naturally, so I believe that it would be necessary to cite (at least some of) the following publications instead: a. J. Bylander, et al., npj Quantum Info. 5, 54 (2019) b. M. Papic, et al., arXiv: arXiv:2305.08916 (2023) c. E. Paladino, et al., Rev. Mod. Phys. 86, 361 (2014) d. G. A. L. White, et al., Nat. Comm. 11, 6301 (2020)
  5. The dashed lines in Fig. 2 cannot be discerned.

  • validity: good
  • significance: good
  • originality: high
  • clarity: high
  • formatting: good
  • grammar: perfect

Author:  Orazio Scarlatella  on 2023-11-14  [id 4111]

(in reply to Report 1 on 2023-06-29)
Category:
answer to question

We thank the Referee for their time spent reviewing our manuscript. We are glad that they appreciated the novelty and accessibility of our manuscript.

The Referee pointed out some questions and requests to be addressed before publication. Here we comment on their questions and requests (following their numbering), pointing out how we accordingly improved our manuscript.

“1. In order to support the authors’ claim, beyond qualitative arguments, that the presented method can be used in the stronger coupling regime, I would suggest that a plot of the difference (perhaps in terms of the process fidelity or similar distance measure) between the NCA and OCA maps (as well as Born – OCA and Born-Markov - OCA) versus the coupling strength of the model is added. This would allow the reader to see: (a) Up to which coupling strength can we expect quantitatively reliable results, (b) how much of an improvement do we see compared to the standard approximations. It would be even more interesting to see a comparison to numerically exact results, however I believe that the OCA approximation can serve as a sufficient guideline for that, up to a certain coupling strength.”

We added the requested plot as 2 additional panels in Fig. 3 and we accordingly significantly rephased section 3.2. We calculated the Jamiolkowski process fidelity (Phys. Rev. A 71, 062310 (2005)) of the NCA(-Markov) maps, and of Born(-Markov) equations, compared to OCA, confirming our claim that the NCA maps can be quantitatively accurate up to quite large couplings, and are significantly more accurate than Born(-Markov) master equations. Note that the plotted fidelity is the minimum fidelity over the time-interval identified by the time-axis of the left panels (a worst-case measure). Note also that in the Ohmic case (top panel) we plotted for values of the coupling up to alpha ~ 0.5, were fidelity starts to deteriorate getting closer to the localization transition, while we plot Born-Markov points only up to alpha ~ 0.2 where it diverges. Instead for the sub-Ohmic case (bottom panel) we plotted up to alpha ~ 0.1, which corresponds to the localization transition in this case [132]. Note that we cannot cross such a point with the present formulation of the maps (but they could be extended) and we now state this more clearly in the manuscript, towards the end of Sec. 3.1.

“2. Furthermore, while the authors claim that the method is general, in the rigorous derivation it is assumed that the bath operators commute under time ordering and later Wick’s theorem for real bosonic variables is applied. In order to claim generality it is necessary to comment on how these steps can be generalized to fermionic or spin environments. If this is not possible it should be mentioned in the main text.”

We rephrased the beginning of Sec. 2, to clarify our assumptions and highlight their generality. We also added a comment in Sec 2.2, that we report here (see manuscript for references): the NCA maps and their higher-order counterparts naturally generalize to fermionic and environments and multiple environments [73], to non-stationary environments [101] and to driven systems. Instead, their derivation doesn’t generalize to spin environments, for which the Wick’s theorem does not apply. Nevertheless, they might provide good approximations also in this case, for weak couplings and low temperatures such that there is few excitations in the environment that effectively behave like bosons.

“3. Some commentary on the feasibility of going to higher order approximations beyond NCA and OCA would be interesting to see.”

We added a comment along with references in Sec 2.2 on the validity of our method. We report it here (see manuscript for references): going one order beyond OCA is feasible [94,106] and a Monte-Carlo sampling around NCA is also possible [75, 105]. Note that, in the former references, the NCA, OCA and “third order” approximations are done for different quantities, still the “third order” approximation is in reach also in our case.

“4. In the introduction, the authors mention that non-Markovian approaches are needed to describe the dynamics of superconducting circuits (SC). One of the three references [47-49] they have cited has nothing to do with SCs, while the other two are implementations of the spin-boson model with SC circuits. From this it is hard to argue that there is really a need to describe the dynamics of these systems beyond the Markovian regime. However, non-Markovian effects are also present in such systems naturally, so I believe that it would be necessary to cite (at least some of) the following publications instead: a. J. Bylander, et al., npj Quantum Info. 5, 54 (2019) b. M. Papic, et al., arXiv: arXiv:2305.08916 (2023) c. E. Paladino, et al., Rev. Mod. Phys. 86, 361 (2014) d. G. A. L. White, et al., Nat. Comm. 11, 6301 (2020).”

We thank the Referee for pointing this issue out, and we apologize for the out-of-context reference. We replaced the references on superconducting circuits with the more pertinent references proposed, to which we added another recent reference.

“5. The dashed lines in Fig. 2 cannot be discerned.”

Here we are not sure what issue the Referee is referring to. On our laptops the older version of Fig. 2 displays correctly (we run Windows and tried Acrobat Reader, Chrome, Microsoft Edge and Firefox), therefore we believe the problem might depend on a particular choice of pdf editor.
Anyway, we generally improved the graphics of Fig. 2 as suggested by Referee 2, therefore we hope the problem has been solved in the process.

---

## Round 5 · Referee Report · Anonymous (Referee 2) · 2023-8-1

Strengths

1- Clear, well-structured, and well-written paper 2- New method for dealing with open quantum system dynamics which can describe physics beyond the standard Born-Markov approximation (quantitively or qualitatively depending on the coupling strength). 3- Methods to numerically solve the open quantum system dynamics with the NCA approximation are similar and have the same structure as the standard Born-Markov theory. 4- The method makes it relatively straightforward to compare cases with and without the Markov approximation, and thereby determine the relevance of the bath memory or of the "non-Markovianity" on specific problems. 5- The validity of the NCA approximation can self-consistently be determined by comparing it to the OCA approximation, which is solved with similar methods. 6- The example of the spin-boson model is well-chosen to illustrate the main features of the proposed NCA approximation method.

Weaknesses

1- Extra memory cost of solving for the dynamical map V (~N^4) instead of the density matrix rho (~N^2) can be very limiting when solving an open quantum system of moderate to large dimension N. 2- Although it is shown that the NCA method goes beyond the Born-Markov (sometimes qualitatively, sometimes quantitatively) this range of extra applicability is still quite limited in practice, as the NCA also requires a type of weak-to-intermediate system-bath coupling regime.

Report

The authors develop the non-crossing approximation (NCA) method for describing the dynamics of open quantum systems, which allows one to go beyond the standard Born-Markov master equation. This is based on solving an equation for the dynamical V(t), which does not require a perturbative second-order approximation in the system-bath coupling (Born-approximation) and also does not require a time-local approximation of dynamics (Markov approximation). Using the example of the well-known spin-boson model at zero temperature, the authors show that the NCA method can qualitatively describe the crossover to the incoherent regime and the onset of the localization quantum phase transition, for which the standard Born-Markov treatment fails even qualitatively. Moreover, in the subohmic case, where interesting features appear at low coupling, NCA shows quantitative validity whereas Born-Markov shows significant deviations (although it is qualitatively valid).

The paper is very clearly written and well-structured. The method is clearly explained using both intuitive arguments and rigorous diagrammatic arguments in the appendix. The validity of the method can also be self-consistently checked by comparing it to the one-crossing approximation (OCA), which is very important to trust in the method when solving problems that are not a priori known.

In general, I think the NCA method can give valuable new insight into the theory of open quantum systems, especially if it could be applied to study non-Markovian and moderate coupling open quantum many-body systems. Nevertheless, I see two strong weaknesses of the method which could limit its practical applicability to describe many-body open quantum systems : 1) The NCA method still requires the assumption of a weak-to-intermediate system bath coupling regime which limits the applicability of the method to ultra-strong coupling cases. This can be already seen in the analysis of the spin-boson model given by the authors, which for strong coupling gives results that are qualitatively correct but not quantitatively. 2) Solving for V(t) instead of rho(t) increases the scaling of memory cost from N^2 to N^4, with N as the dimension of the open quantum system. I think this can be very limiting even for moderate system sizes of a few spins and I do not agree with the statements of the authors that this is a little extra numerical cost. In my experience, even the scaling N^2 of the standard Born-Markov Master equation strongly limits the size of the system under study compared to the scaling of standard quantum trajectory calculations which grows with N when describing the dynamics of a single trajectory.

The paper is of very good quality but before judging if it is suitable for publication in SciPost or not, I would like the authors to provide some further information so that I can better evaluate the practical applicability and impact of the method.

1) Potential for discovering qualitatively new physics in open quantum systems: As far as I understand the NCA assumes that the state of the system evolves as rho(t) = V(t) rho(0). Why this does not imply that evolution is local in time? Which kind of non-Markovian effects can be accounted for by the NCA method? This is not so much discussed in the work and would be very interesting to comment/develop it a bit. For instance, interesting non-Markovian physics that does not require strong coupling are retardation effects in the interaction mediated by two or more emitters coupled to a common waveguide. Could this be described by the NCA or is this out of the approximation similar to the Born-Markov approximation? I would strongly appreciate it if the authors could answer these questions and also comment on the prospects and limitations for studying retardation or memory effects in super- and sub-radiant systems of many emitters coupled to a common waveguide.

2) What are the memory and time costs of evaluating the OCA? I see that also solving this is crucial to validate the numerical results of the NCA and may be always needed when studying an unknown open quantum system. This can further increase the numerical cost as well.

3) With the aim of applying the NCA method to large systems, a possible solution to reduce the memory overhead of the scaling N^4 could be using more memory-efficient methods to store the matrices such as matrix product operators (MPO). A short discussion in the outlook discussing ways to apply the NCA method to larger systems would be very instructive and would highly improve the impact/applicability of the work I think.

4) The fact that the quantum fluctuation regression theorem is also valid in the case of the more general NCA approximation adds a lot of value to the method and it is a pity that this is not so much discussed in the text. Being able to easily access two-time correlation functions of open quantum systems can be very useful and I suggest the authors expand this discussion in the text. Commenting, for instance, how the numerical costs of these calculations scale with system size and time, and also which kind of correlations can these methods describe that are not included in the standard Born-Markov procedure.

Minor comments: 1) In abstract, intro, and conclusions: As discussed above, I do not agree with the statement that the NCA only requires “very little extra numerical cost”. This is only true for small systems of one or two spins. I suggest the authors reformulate this statement to something like "at the expense of increasing numerical cost" or similar.

2) In Figure 2 (top): Use the same x-axis on the left and right panels as these will make the comparison easier between NCA and standard Born- and Born-Markov approaches.

3) In all figures, you could also put small titles to identify better the subplots. For instance, in Figure 8 you could put NCA, and BM on the left of the panels, and above the panels you could put “under-damped coherent” and “incoherent”. Then it is much easier to visually interpret and compare the plots.

4) Final sentence at the bottom of page 19 is not very clear to me. I do not understand what is exactly not well reproduced by the NCA method. It would be nice if the authors could specify this a bit better.

Requested changes

1- Authors could comment on the prospects of combining the NCA method with matrix product operators (MPO) as this could reduce the memory cost when dealing with large open quantum systems, at least in 1D. 2- Authors could expand a bit the outlook, commenting on possibilities that this method opens for describing non-Markovian open quantum systems, for instance regarding long time delays when the coupling is not that high (as for instance in the case of many emitters coupled to a common waveguide), as this could be a regime where NCA can be much better than any other Born- or Born-Markov method. For instance, super- and subradiance effects could be precisely described if the limitation in Hilbert space dimension is mitigated via MPOs or in cases when the system size is not that large.

  • validity: good
  • significance: good
  • originality: high
  • clarity: top
  • formatting: excellent
  • grammar: excellent

Author:  Orazio Scarlatella  on 2023-11-14  [id 4110]

(in reply to Report 2 on 2023-08-01)
Category:
answer to question

We thank the Referee for their time spent reviewing our manuscript. We are glad that they appreciated the novelty and strengths of our approach and found our manuscript clear.

The Referee asked for further information regarding the practical applicability and impact of the method before expressing his opinion. Here we comment on their questions and requests, following their numbering, pointing out how we improved our manuscript accordingly.

“1) Potential for discovering qualitatively new physics in open quantum systems: As far as I understand the NCA assumes that the state of the system evolves as rho(t) = V(t) rho(0). Why this does not imply that evolution is local in time? Which kind of non-Markovian effects can be accounted for by the NCA method? This is not so much discussed in the work and would be very interesting to comment/develop it a bit. For instance, interesting non-Markovian physics that does not require strong coupling are retardation effects in the interaction mediated by two or more emitters coupled to a common waveguide. Could this be described by the NCA or is this out of the approximation similar to the Born-Markov approximation? I would strongly appreciate it if the authors could answer these questions and also comment on the prospects and limitations for studying retardation or memory effects in super- and sub-radiant systems of many emitters coupled to a common waveguide.”

We thank the Referee for their question that allowed us to clarify these important aspects.
We realize that we improperly used the notion of “local in time” dynamics, as we discuss below, therefore we now more properly phrase the discussion in terms of non-Markovian effects. First, we note that it is formally always possible to define a map V(t,t’), that maps the system density matrix from earlier to later times rho(t)=V(t,t’) rho(t'). Importantly, V(t,t’) might depend on rho(t_1) at times earlier than t, therefore describing memory effects and leading to a non-Markovian dynamics. Whether this is the case, it is clear from its equation of motion. In our case, V(t,t’) obeys Eq. (4) that is an integro-differential equation, which couples it explicitly to its values at earlier times, making memory effects manifest. Note that in Eq. (4) we set the time t’=0, assuming that system and bath states are uncorrelated at that time (only), and that the bath is stationary, therefore V(t,t’) reduces to V(t). In particular, coupling to earlier times is both due to the time-integral in Eq. (4), and to the fact that the dissipator depends on the map at previous times. Indeed, these are the two main sources of non-Markovianity in the NCA maps. Note that in the NCA-Markov map the former mechanism is removed, but not the latter, allowing it to still capture serious non-Markovian effects. To clarify this aspect, we now specified that this map is obtained by a partial, rather than full, Markovian approximation.
On the contrary, an evolution equation that is strictly local in time, would only couple V(t) with itself and its derivatives at the same time corresponding to a differential equation, and would have no retardation and a non-self-consistent dissipator. This is the case of a fully Markovian dynamics, described for example by a Lindblad master equation. It is not the case instead of Eq. (6) for the NCA-Markov map, because of the self-consistent dissipator, so we now avoid calling it “time-local”. The reason for this previous improper denomination is that the approximation does allow to reduce the numerical complexity to that of a time-local equation, in a way that we now discuss in the manuscript.
We clarified these aspects mainly in Sec. 2.1 describing the NCA-Markov map, but also after Eq. (4) and Eq. (5) for the NCA map.
We also thank the referee for pointing out the interesting problems of multiple emitters coupled to a common waveguide, that can display super and sub-radiance. Indeed, this is an excellent example of where our maps might make a difference in capture retarded, coherent and dissipative effects, mediated by the reservoir. We now mention this in the conclusion, where we added a reference to a recent review article.

“2) What are the memory and time costs of evaluating the OCA? I see that also solving this is crucial to validate the numerical results of the NCA and may be always needed when studying an unknown open quantum system. This can further increase the numerical cost as well.”

We added a comment in Sec 2.3 on the numerical implementation. We report it here (see manuscript for references): in the case of OCA the cost gets an additional O(t^2) factor, owing to the additional time-integrals involved (thus O(t^4) keeping all memory effects, or O(t^3) with a Markovian approximation). The downside is a longer computational time, especially for long-times calculations. Note instead that the size of the map V(t) doesn’t change, therefore this is not a limitation in the system Hilbert-space size. We also remark that computing OCA might be necessary for large system-bath couplings, while for small couplings OCA contributions are suppressed as 1/lambda^2 (with lambda the system-bath coupling strength), thus one can trust NCA results without the compelling need of computing OCA corrections.

“3) With the aim of applying the NCA method to large systems, a possible solution to reduce the memory overhead of the scaling N^4 could be using more memory-efficient methods to store the matrices such as matrix product operators (MPO). A short discussion in the outlook discussing ways to apply the NCA method to larger systems would be very instructive and would highly improve the impact/applicability of the work I think.”

We thank the Referee for suggesting mentioning explicitly that compression with MPOs methods is possible, as a large community is working to optimize those methods. We added a comment both in the Conclusion and in Sec 2.3 on the Numerical Implementation.

“ 4) The fact that the quantum fluctuation regression theorem is also valid in the case of the more general NCA approximation adds a lot of value to the method and it is a pity that this is not so much discussed in the text. Being able to easily access two-time correlation functions of open quantum systems can be very useful and I suggest the authors expand this discussion in the text. Commenting, for instance, how the numerical costs of these calculations scale with system size and time, and also which kind of correlations can these methods describe that are not included in the standard Born-Markov procedure.”

We moved the discussion of the quantum regression theorem (and the steady-state equation) in the main text to give it more emphasis: it is now in Sec. 2.1. It was previously in the appendix, mainly because its application to the calculations correlation functions of the spin-boson model was also in the appendix, but we agree that it is a remarkable result of general interest. We now present the result in a more general terms and we pointed out that the numerical cost is the same as for computing the dynamics.
The correlations captured are the same entering the map V(t), that allows to compute both single-time averages and correlation functions: whenever the OCA corrections are negligible, and note that this can be checked for the entire map V(t) rather than for single-time averages only, also the correlation functions will be.

The Referee also identified some “Minor comments” that we took into account in the updated manuscript and we address in the following.

“1) In abstract, intro, and conclusions: As discussed above, I do not agree with the statement that the NCA only requires “very little extra numerical cost”. This is only true for small systems of one or two spins. I suggest the authors reformulate this statement to something like "at the expense of increasing numerical cost" or similar.”

We agree that those statements were misleading and rephrased them.
We previously meant to highlight that the cost for time propagation is similar to that of master equations: of order t^2(t) for NCA(-Markov), as for Born(-Markov) approaches, allowing to propagate up to long times. Note that this cost increases going to higher-order approaches (like OCA), because of additional time-integrals involved. In any case, we agree with the Referee that the statement was imprecise, as the scaling with the system Hilbert space size is indeed worse than for master equations (O(N^4) instead of O(N^2)). Anyway, we believe this is still a moderate scaling compared to numerically exact methods for open quantum systems, which in the worst case have an exponential scaling. Therefore, we replaced the statements of “very little extra numerical cost” with “moderate extra numerical cost”, which we hope the Referee will find adequate.

“2) In Figure 2 (top): Use the same x-axis on the left and right panels as these will make the comparison easier between NCA and standard Born- and Born-Markov approaches.”

We now use the same axes. We also generally improved Fig. 2.

“3) In all figures, you could also put small titles to identify better the subplots. For instance, in Figure 8 you could put NCA, and BM on the left of the panels, and above the panels you could put “under-damped coherent” and “incoherent”. Then it is much easier to visually interpret and compare the plots.”

We changed Fig. 2 and Fig. 8 to reflect this suggestion.

“4) Final sentence at the bottom of page 19 is not very clear to me. I do not understand what is exactly not well reproduced by the NCA method. It would be nice if the authors could specify this a bit better.”

We clarified this sentence. We report it here (see manuscript for references): the exact functional dependence of delta_r on alpha close to the critical point delta_r = c delta(delta/omega_c)^{alpha/(1-alpha)} [119,146] (where c is a constant prefactor) is not correctly reproduced by the NCA.

---

## Round 6 · Referee Report · Anonymous (Referee 2) · 2023-12-7

Report

I thank the authors for taking my comments very seriously , and for making the changes/adaptations in the paper to clarify better the points I raised. All my questions are properly discussed and clarified. I think the paper is of high quality and the method for solving non-Markovian quantum system is quite original, so it can be a very valuable contribution to the community. I recommend publication of this paper in SciPost as is.

---

## Round 6 · Referee Report · Anonymous (Referee 1) · 2023-12-26

Report

I would like to thank the authors for addressing my questions. All of the additional information that I have requested has been added and I can now confidently approve the revised manuscript for publication in SciPost.

---

## Round 6 · List of Changes

- We clarified the assumptions and generality of our approach, in response to Report 1, at the beginning of Sec. 2 and at the end of Sec. 2.2.
- We commented and clarified on the non-Markovian effects captured by our approach, below Eqs. (4), (5) and mainly below Eq. (6), in response to Report 2.
- We moved the discussion of the quantum regression theorem (and the steady-state equation) from the Appendix into the main text in Sec. 2.1, to give it more emphasis as suggested by Report 1. We also added a comment on the numerical cost of correlation functions calculations.
- We commented on the feasibility of going beyond NCA in Sec. 2.2, in response to Report 1.
- We added a comment on the computational cost of OCA in Sec. 2.3, in response to Report 2.
- We changed the wording that our approach sees a “very little increase” in numerical cost compared to standard master equation, with “moderate increase”, throughout the manuscript, in response to Report 2.
- We commented in Sec. 2.3 and in the Conclusions about the possibility of compressing our maps with matrix product operators, in response to Report 2.
- We improved the graphics of Fig. 2 in the main text and Fig. 8 in the supplemental in response to Report 2.
- We added 2 panels in Fig.3 showing the process fidelity of the various maps, as compared to the most accurate OCA approximation, as a quantitative benchmark of accuracy, in response to Report 1. We also rephrased Sec. 3.2 accordingly.
- We clarified the statement with references [119,146] in App. E.1 as suggested by Report 2.

---

## Editorial Decision

published